# Leveraging Semantic and Positional Uncertainty for Trajectory Prediction

## Abstract

Given a time horizon with historical movement data and environmental context, trajectory prediction aims to forecast the future motion of dynamic entities, such as vehicles and pedestrians. A key challenge in this task arises from the dynamic and noisy nature of real-time maps. This noise primarily stems from two resources: (1) positional errors due to sensor inaccuracies or environmental occlusions, and (2) cognitive errors resulting from incorrect scene understanding. In an attempt to solve this problem, we propose a new framework that estimates two kinds of uncertainty, *i.e.*, positional uncertainty and semantic uncertainty simultaneously, and explicitly incorporates both uncertainties into the trajectory prediction process. In particular, we introduce a dual-head structure to independently perform semantic prediction twice and positional prediction twice, and further extract the prediction variance as the uncertainty indicator in an end-to-end manner. The uncertainty is then directly concatenated with the semantic and positional predictions to enhance the trajectory estimation. To validate the effectiveness of our uncertainty-aware approach, we evaluate it on the real-world driving dataset, *i.e.*, nuScenes. Extensive experiments on 4 mapping estimation and 2 trajectory approaches show that the proposed method (1) effectively captures map noise through both positional and semantic uncertainties, and (2) seamlessly integrates and enhances existing trajectory prediction methods on multiple evaluation metrics, *i.e.*, minADE, minFDE, and MR.

## 1 Introduction

Accurate and efficient prediction of future vehicle trajectories is a critical task in autonomous driving systems (Zhou et al., 2022; Gu et al., 2021; Ngiam et al., 2022; Wu et al., 2023). To generate reliable trajectory predictions, autonomous vehicles should thoroughly understand and process the surrounding environment. High-Definition (HD) maps are essential for this task. However, the dynamic nature of the environment poses significant challenges to accurate trajectory prediction. For example, pedestrians may suddenly enter the path of vehicle, weather and visibility conditions can fluctuate, obstacles may obstruct the view, and sensor errors can introduce noise. These factors can lead to discrepancies in the vehicle perception of map information, thereby affecting the performance of trajectory prediction.

The existing trajectory prediction works concentrate on two key aspects. **(1) One line of works focuses on the High-Definition (HD) maps estimation.** The early works usually construct HD maps in an offline process, which heavily relies on SLAM (Simultaneous Localization and Mapping) techniques (Shan & Englot, 2018; Zhang & Singh, 2014). However, SLAM usually requires extra maintenance costs. Therefore, some researchers resort to the bird's-eye view (BEV) representations (Chen et al., 2022; Li et al., 2022b; Zhou & Krähenbühl, 2022; Hu et al., 2021), which uses deep neural networks to extract and fuse map information from multiple sensors and environmental data in an end-to-end manner. However, such methods typically do not provide a vectorized path, which represents the road as a sequence of interconnected keypoints. This representation allows for a more precise depiction of the road's geometric and topological characteristics. To further enhance the expressiveness of the map, some approaches (Li et al., 2022a; Liu et al., 2023; Liao et al., 2023a;b; Xu et al., 2024; Li et al., 2024) have adopted a vectorized map format. This format not only preserves detailed environmental information but also aligns more closely with the structure of trajectory data, thereby facilitating downstream tasks such as path planning and trajectory prediction.

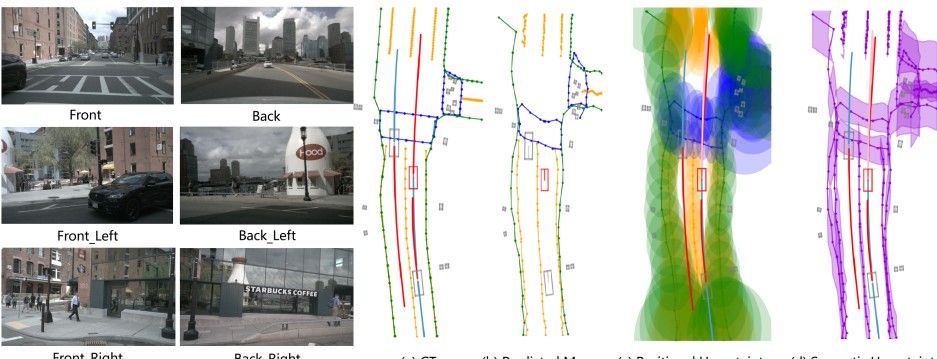

Figure 1: **Motivation.** The 6 images on the left are captured by 6 different cameras on the vehicle. The map estimation remains challenging from RGB images, and thus inevitably contain noise, accumulating the error to the trajectory prediction. Comparing ground-truth high-definition (HD) map (a) and the predicted map in (b), we could see the error usually occurs in the uncertain areas. Therefore, in this work, we intend to leverage two types of uncertainty, *i.e.*, positional uncertainty and semantic uncertainty, to indicate the map errors, mitigating the negative impacts. (c) shows the positional uncertainty for three categories shown in three colors: green for boundary, blue for pedestrian crossing, orange for divider, and red for the ego car. The greater the positional uncertainty of the three categories, the larger the ellipse centered on the map element. (d) shows the semantic uncertainty of our constructed high-definition map, where the purple error band indicates the likelihood of being misclassified as another category.

**(2) Another line of works focuses on directly refining the trajectory prediction model.** Some pioneering works (Cui et al., 2019; Jain et al., 2019; Chai et al., 2019; Liang et al., 2020a) usually extract rasterized BEV features from image inputs via Convolutional Neural Networks (CNNs), while recent works apply transformers (Vaswani et al., 2017; Zhou et al., 2022) or GNNs (Gao et al., 2020; Liang et al., 2020b; Zeng et al., 2021; Zhao et al., 2020) to capture the relationships within the vectorized map. However, both lines of works suffer from the inherent data noise, such as occlusions, weather changes, and other environmental complexities (see Figure 1 left), and have not explicitly conducted the noise modeling. As shown in Figure 1 (a) and (b), map estimation inevitably contains the noise. This leads to error accumulation during the trajectory prediction training process, ultimately affecting the final performance.

Therefore, in this work, we intend to explicitly model noise during training and regularize the training process. **It is worth noting we do not remove the noise, but mitigate the negative impact of such noises**. Specifically, we consider uncertainty in prediction and illustrate the relationship between noise and uncertainty in Figure 1 (c,d). We observe that high noise in the input data leads to greater uncertainty in map estimation. To describe noise more precisely, we categorize it into two types: (1) noise that causes positional errors, such as sensor inaccuracies or environmental occlusions (Figure 1 (c)), and (2) noise that causes cognitive errors due to incorrect scene understanding (Figure 1 (d)). These are modeled as positional uncertainty and semantic uncertainty, respectively. In the implementation, we introduce a dual-head structure. The primary head gets features from "res5c" and the auxiliary head gets features from "res4f". "res5c" and "res4f" are commonly used layer names in the ResNet50 backbone. "res5c" corresponds to the output of the final layer in the last block of ResNet50, whereas "res4f" refers to the output of the last layer in the block preceding "res5c". Both of these two heads are used to regress the semantic information and positional information. The difference between the two heads is a measure of uncertainty in semantic and positional information. For either semantic prediction or positional prediction, our model independently performs twice and two groups of results. We further compute the prediction variance as the uncertainty indicator. Then the map elements, enriched with positional and semantic uncertainties, are fed into downstream trajectory prediction models. This enables the model to leverage the uncertainty context of map elements, leading to more accurate trajectory predictions. In summary, our contributions are as follows:

- We observe an inherent problem in map estimation for trajectory prediction, *i.e.*, the presence of noise in High-Definition (HD) maps. While it is impractical to eliminate this noise entirely, we propose a new approach that leverages two types of uncertainty, *i.e.*, positional and semantic, to indicate and mitigate its negative impacts. By explicitly integrating these

uncertainties as noise indicators into the model training process, our method effectively reduces the adverse effects of data noise, thereby enhancing the robustness and accuracy of trajectory predictions.

- Albiet simple, our approach can be seamlessly integrated with 4 exisitng mapping estimation and 2 trajectory approaches, to consistently improves their prediction accuracy. For instance, when incorporating MapTRv2-Centerline as map backbone and HiVT as trajectory prediction backbone, we further improve minADE by 8%, minFDE by 10%, and MR by 22% on the nuScenes dataset.

## 2 RELATED WORK

**Map-Informed Trajectory Prediction.** Map-based trajectory prediction is closely tied to advancements in map estimation. Current vectorized methods can be broadly divided into two two categories: one is using graph neural networks (GNNs) (Gao et al., 2020; Liang et al., 2020b; Zeng et al., 2021; Zhao et al., 2020) another is leveraging transformers (Vaswani et al., 2017) with cross-attention mechanisms (Vaswani et al., 2017; Deo et al., 2021; Gu et al., 2021; Liu et al., 2021; Zhou et al., 2022; Gu et al., 2024). GNN-based methods use graph networks to extract entity features and model interactions between different entities. LaneGCN (Liang et al., 2020b) constructs a lane graph and applies multiple adjacency matrices and extended graph convolutions along lane expansions to capture the complex topology of the lane graph. LaneRCNN (Zeng et al., 2021) proposes a local lane graph representation (LaneRoI) for each agent to encode its past motion and local map topology, modeling agent interactions through graph-to-graph interactions. On the other hand, transformer-based methods with cross-attention mechanisms have become the most widely used and state-of-the-art approaches. These methods employ cross-attention between map elements and agents to achieve high-performance predictions. Zhou et al. (2022) proposes the Hierarchical Vector Transformer method, which extracts local context and models global interactions, enabling more robust multi-agent motion prediction. Recently, Gu et al. (2024) expose the uncertainty of map element regression and classification to downstream behavior prediction tasks. TopoNet Li et al. (2023) directly infers the connectivity between lane centerlines and various traffic elements from sensor inputs. Gu et al. (2025) propose exposing the rich internal features of online map estimation methods by utilizing the abundant intermediate features generated during the PV2BEV conversion from the encoder's perspective view to a bird's-eye view. These rich internal features generated during HD map estimation are also leveraged during the prediction phase, using internal BEV features to enhance performance. Although the approach proposed by Gu et al. (2024) introduces uncertainty representation in vectorized HD maps, the predicted uncertainty is incomplete and does not fully address the noise present in vectorized HD maps. Different from the Gu et al. (2024) approach, our work defines two types of uncertainty in map-based trajectory prediction tasks. The accuracy of the trajectory prediction task is enhanced by addressing the issue of map noise through positional and semantic uncertainty.

**Online Map Estimation.** Online map estimation leverages onboard sensors, environmental data, and vehicle trajectories to dynamically update and optimize map information in real time, ensuring accuracy and adaptability in changing environments. Existing approaches for online map estimation can be broadly categorized into two types: rasterized encoding and vectorized encoding. Rasterized encoding methods (Chai et al., 2019; Cui et al., 2019; Liang et al., 2020a; Casas et al., 2018) primarily use a 2D bird-eye view (BEV) perspective, projecting and fusing 3D data to generate rasterized semantic segmentation representations of the static world, typically encoded through CNNs. For instance, Casas et al. (2018) develops a CNN-based detector and predictor to process 3D point clouds from LiDAR sensors and dynamic maps of the environment. However, the grid-based nature of convolutions in these methods limits model ability to capture fine structural details of high-definition maps, as non-grid sampling is not possible. To overcome the drawbacks, vectorized encoding methods have gradually replaced traditional rasterized BEV approaches. These methods (Liu et al., 2022; Philion & Fidler, 2020; Li et al., 2022c; Liang et al., 2020a), utilizing encoder-decoder architectures, directly regress and classify map elements such as polylines and polygons, improving adaptability and accuracy in dynamic scenarios. For example, HDMapNet (Li et al., 2022a) and SuperFusion (Dong et al., 2022) fuse image data from surround-view cameras and point cloud features from LiDAR into BEV representations, which are then processed to extract vectorized map elements. Moreover, the MapTR series(Liao et al., 2023a;b; Xu et al., 2024) of works build a structured, par-

allel, single-stage framework, framing vectorized HD map estimation as a point-set prediction task, significantly improving estimation efficiency. StreamMapNet (Yuan et al., 2024) further introduces multi-attention and temporal information to incorporate frame-level temporal data, providing high stability for large-scale local HD maps. MapTracker (Chen et al., 2025) introduces the concept of HD mapping as tracking and utilizes the history of memory latent in BEV and Vector representations to achieve temporal consistency. MGMap (Liu et al., 2024) proposes using learned masks through a mask-guided strategy to enhance instance-level features with global and structural information and refined point-level information through mask patches, enabling more precise map feature localization on bird's-eye view feature maps of different scales. MapDistill (Hao et al., 2025) employs the knowledge distillation (KD) approach for efficient high-definition map construction by transferring a model that fuses camera and LiDAR information into a lightweight pure camera model. Additionally, a high-efficiency transfer module was designed to enhance the student model's feature representation for HD map construction. Despite these advancements, none of these methods address the noise inherent in online map estimation. To tackle this, our approach introduces uncertainty modeling to enhance online map estimation accuracy in noisy environments.

**Uncertainty Learning.** Uncertainty learning gains significant attention in the fields of trajectory prediction and map estimation for autonomous driving, as managing uncertainty is crucial for making reliable and safe predictions in dynamic and noisy environments. For instance, Ma et al. (2019) proposes an LSTM-based real-time traffic prediction algorithm, improving prediction by learning agent movement and categories through an instance layer and a category layer, respectively. Generative models, such as GANs (Lv et al., 2022), also capture behavioral variability effectively. Zhou et al. (2022) obtains each agent position at each time step in the local coordinate system while using an MLP to estimate its corresponding uncertainty, incorporating trajectory uncertainty into the regression loss. Recent methods introduce uncertainty-aware models to apply map-derived uncertainties to downstream trajectory prediction tasks. Gu et al. (2024) exposes map element uncertainty to downstream trajectory prediction, enhancing prediction reliability in noisy environments. However, the uncertainty in Gu et al. (2024) remains incomplete, as it is derived through linear regression layers without detailed analysis of map positional uncertainty or model error in scene understanding. To address this, we propose estimating two levels of uncertainty. By passing both positional and semantic uncertainties from the online vectorized map estimation process into downstream prediction tasks, our approach enhances the informative value of maps for prediction tasks, resulting in more accurate and reliable predictions in real-world driving scenarios. This method allows the model to better handle dynamic environments, sensor noise, and occlusions, achieving superior performance in prediction accuracy and robustness compared to existing approaches.

## 3 METHOD

### 3.1 UNCERTAINTY ESTIMATION

We show the brief trajectory prediction pipeline in Figure 2. We extract 2D features from the vehicle camera images and transform them into BEV (Bird's-Eye View) features. To capture positional and semantic uncertainty, we introduce a dual-head structure consisting of a primary head and an auxiliary head with identical structure. For each BEV feature, we perform two predictions using primary and auxiliary heads. Each head outputs a set of positional and semantic predictions. We then conduct location regression and semantic regression on the features from both the primary and auxiliary heads. The primary and auxiliary location information are used to compute the KL divergence, which serves as the positional uncertainty. Before feeding this information into the downstream trajectory prediction task, we calculate the mean and MSE to obtain the mean semantic information and the semantic uncertainty. The high-definition map location information, semantic information, and their corresponding uncertainties, obtained through our uncertainty estimation, are integrated into the representation of the encoded map in the downstream prediction model. Next, we will elaborate the details.

**Positional Uncertainty.** In particular, to estimate the position of map elements, we first adopt an MLP-based structure to regress a two-dimensional vector representing the normalized BEV coordinates $(x, y)$ of each map element. We then design an auxiliary head with a structure similar to the primary head. The only difference is that we additionally introduce one dropout layer to increase the variability in prediction. Thus, for each map element, we obtain the primary map element vector $\boldsymbol{\mu}$ and the auxiliary map element vector $\boldsymbol{\mu}'$. Following Gu et al. (2024), we apply the Laplace

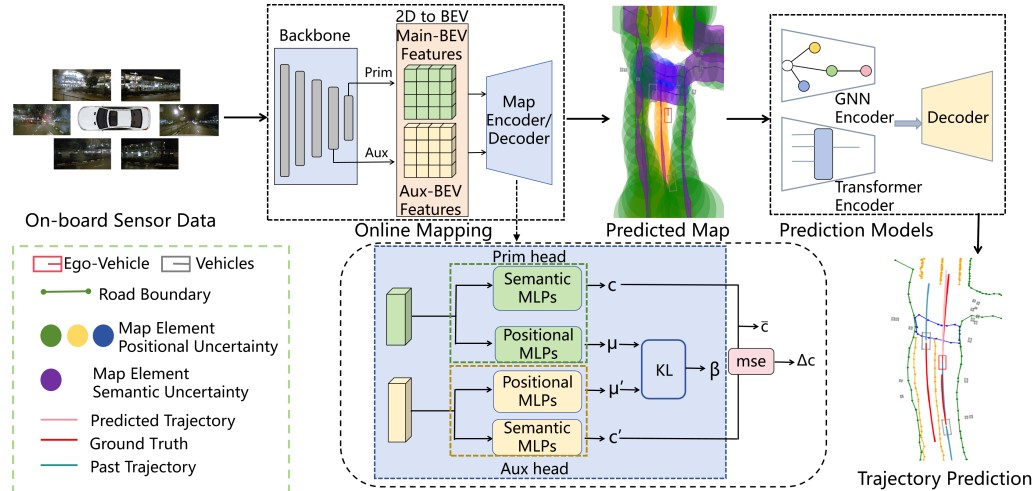

Figure 2: **Overall pipeline.** Firstly, given image from vehicle camera, we extract the 2d features and transform them to bev feature. for every bev feature we then predict the BEV feature twice via primary and auxilary head. Secondly, in the uncertainty estimation, during the map estimation stage, we perform location regression and semantic regression on the features of both the primary and auxiliary heads. The resulting primary and auxiliary location information are used to calculate KL divergence, with the output serving as positional uncertainty, denotes as $\boldsymbol{\mu}$ and $\beta$. The primary and auxiliary semantic information are retained, and before inputting to the downstream trajectory task. We obtain the mean semantic information $\bar{c}$ and semantic uncertainty $\Delta c$. Thirdly, we concatenate the high-definition map location information, semantic information, and their uncertainties, as the input of the downstream model to enhance the scene understanding for trajectory prediction.

distribution to both $\boldsymbol{\mu}$ and $\boldsymbol{\mu}'$. To better estimate positional uncertainty, we calculate the KL divergence between $\boldsymbol{\mu}$ and $\boldsymbol{\mu}'$ and use that to quantify positional uncertainty for each map element. Mathematically, this process is defined as:

$$\beta = \mathbb{E}\left[\boldsymbol{\mu} \log\left(\frac{\boldsymbol{\mu}}{\boldsymbol{\mu}'}\right)\right]. \tag{1}$$

If the predicted vectors from the two regression heads diverge significantly, the approximate variance will be large, reflecting the model uncertainty about the prediction. This uncertainty enables a more detailed description of positional noise in each map element and captures the model confidence in its predictions.

**Semantic Uncertainty.** For semantic uncertainty, we also use two heads to process BEV features from the input, each independently producing a set of class scores. We denote the classification probability from primary and auxiliary heads as $c$ and $c'$, respectively. For better usage in downstream tasks, we calculate the mean of $c$ and $c'$, denoted as $\bar{c} = \frac{1}{2}(c + c')$, to serve as the updated confidence score for the map element. Meanwhile, we compute the MSE between the confidence scores $c$ and $c'$ from the primary and auxiliary heads, using this divergence as a supplementary uncertainty measure $\Delta c$ for semantic classification confidence:

$$\Delta c = (c - c')^2. \tag{2}$$

Our semantic uncertainty for map elements thus consists of two key components: the mean classification confidence score and the supplementary uncertainty information calculated from the MSE between the classification probability of the two heads.

**Discussion. Why use an auxiliary head to estimate uncertainty?** By introducing an auxiliary head that extracts features from RGB images, the model captures a different receptive field compared to the primary head. While the primary head focuses on deeper-level features, the auxiliary head processes relatively shallower ones. This multi-layered feature extraction ensures that both deep and shallow image features are considered. The variation in feature extraction between the two heads provides valuable insights for uncertainty estimation. Discrepancies in predictions from the main and auxiliary heads help gauge the level of uncertainty. Additionally, when estimating both positional and semantic uncertainty, we introduce a dropout layer after the auxiliary head.

This introduces variability in the positional and semantic features during training, amplifying the differences between the predictions. These enhanced discrepancies improve the model's ability to estimate uncertainty, thereby enhancing the robustness and accuracy of the trajectory predictions. **Why perform positional and semantic uncertainty separately?** The core objective is to enrich map elements with more diverse and accurate information, while simulating real-world conditions such as occlusions and sensor errors, which can affect map prediction accuracy. These factors can lead to imprecise location predictions, resulting in errors in subsequent agent trajectory predictions. Additionally, downstream tasks rely on map elements that contain both positional and semantic information. By separately estimating positional and semantic uncertainties, we provide a more comprehensive representation of the environment. This allows downstream prediction networks to better leverage both spatial positions and their corresponding semantic features, leading to more reliable and robust trajectory predictions. **The compatibility of the proposed uncertainty.** Our uncertainty estimation method is highly compatible with advanced map element estimation approaches. We verify this by integrating our uncertainty estimation into four state-of-the-art online HD mapping methods: MapTR (Liao et al., 2023a), MapTRv2 (Liao et al., 2023b), MapTRv2-Centerline and StreamMapNet (Yuan et al., 2024). Both MapTR (Liao et al., 2023a) and MapTRv2 (Liao et al., 2023b) utilize an encoder-decoder architecture to transform RGB images into BEV (Bird's-Eye View) features using the LSS (Lift, Splat, Shoot) method. When incorporating our proposed uncertainty, we adopt a perception processing method similar to prior work (Gu et al., 2024). This ensures that the four types of map element information generated by these models are constrained within a perception range centered around the autonomous vehicle, with a longitudinal range of 60 meters and a lateral range of 30 meters. This enriched and uncertainty-aware map information enhances the accuracy and robustness of trajectory prediction learning. By providing more comprehensive and reliable map data, our approach enables downstream models to better handle real-world conditions and uncertainties, leading to improved performance in trajectory prediction tasks.

## 3.2 Uncertainty-aware Trajectory Prediction

Trajectory prediction aims to predict the future trajectory of traffic agents in highly dynamic environments. Traditionally, it first encodes vertex coordinates through MLPs within the encoder and then integrates with the GNN or attention layers in Transformers to capture long-term dependencies between entities. Our uncertainty-aware trajectory prediction method specifically incorporates the positional uncertainty and semantic uncertainty introduced in Section 3.1 during the encoder process. Our input for the trajectory prediction consists of four types of uncertainty information: map positional uncertainty $\mu$, differentiable information $\beta$, semantic class probability $\bar{c}$ derived from semantic uncertainty estimation, and supplementary semantic variation $\Delta c$. We combine these four uncertainty representations into a unified encoding and form the uncertainty-aware map information. This process can be formulated as:

$$E_{unc} = \text{MLPs}\left[\text{concat}\left(\mu, \beta, \bar{c}, \Delta c\right)\right], \tag{3}$$

where concat denotes the concatenation operation, $\bar{c}, \Delta c \in \Phi^{C-1}$ represent the probability simplex with $C$ classes. Our uncertainty-aware trajectory prediction method integrates seamlessly with two state-of-the-art vehicle trajectory prediction models: HiVT (Zhou et al., 2022) and DenseTNT (Gu et al., 2021). HiVT is a Transformer-based approach that treats vectorized map elements as a sequence of tokens. In our approach, map elements enriched with positional and semantic uncertainty are input as point sets into the HiVT encoder. The positional and semantic uncertainty will be concatenated and jointly encoded during the local encoding stage. On the other hand, DenseTNT is a GNN-based approach and our map elements with uncertainty information can be directly encoded using the VectorNet (Liu et al., 2023).

**Discussion. What are the advantages of the proposed uncertainty-aware trajectory prediction method?** Accurate vehicle trajectory prediction is highly dependent on high-definition (HD) map data, as map elements are crucial for predicting agent trajectories. While some previous methods Gu et al. (2024) have utilized map uncertainty to enhance trajectory predictions, they often focus solely on Laplace-distributed location uncertainties and provide only basic class probabilities. Different from existing works, our proposed approach incorporates both positional and semantic uncertainties, thereby enriching the map elements with more comprehensive uncertainty information. This enhanced representation allows the prediction model to better leverage contextual information, leading to more accurate and robust trajectory forecasting.

Table 1: Quantitative results of eight experiments combining 4 high-definition map estimation models and 2 trajectory prediction models on the nuScenes (Caesar et al., 2020) dataset are presented. Overall, we observe that our method, which integrates both positional and semantic uncertainties, outperforms previous approaches in enhancing the prediction model performance, with the most significant improvement seen in the MapTRv2-centerline method and StreamMapNet method.

| Online HD Map Method | Prediction Method | | | | | |
| --- | --- | --- | --- | --- | --- | --- |
| | HiVT (Zhou et al., 2022) | | | DenseTNT (Gu et al., 2021) | | |
| | minADE ↓ | minFDE ↓ | MR ↓ | minADE ↓ | minFDE ↓ | MR ↓ |
| MapTR (Liao et al., 2023a) | 0.4015 | 0.8418 | 0.0981 | 1.091 | 2.058 | 0.3543 |
| MapTR (Liao et al., 2023a) + (Gu et al., 2024) | 0.3854 | 0.7909 | 0.0834 | 1.089 | 2.006 | 0.3499 |
| MapTR (Liao et al., 2023a) + Ours | **0.3660** (−5%) | **0.7564** (−5%) | **0.0745** (−11%) | **0.954** (−13%) | **1.909** (−5%) | **0.3429** (−2%) |
| MapTRv2 (Liao et al., 2023b) | 0.4057 | 0.8499 | 0.0992 | 1.214 | 2.312 | 0.4138 |
| MapTRv2 (Liao et al., 2023b) + (Gu et al., 2024) | 0.3930 | 0.8127 | 0.0857 | 1.262 | 2.340 | **0.3912** |
| MapTRv2 (Liao et al., 2023b) + Ours | **0.3697** (−3%) | **0.7621** (−6%) | **0.0787** (−8%) | **1.099** (−13%) | **2.235** (−5%) | 0.4230 (+8%) |
| MapTRv2-Centerline (Liao et al., 2023b) | 0.3790 | 0.7822 | 0.0853 | 0.8466 | 1.345 | 0.1520 |
| MapTRv2-Centerline (Liao et al., 2023b) + (Gu et al., 2024) | 0.3727 | 0.7492 | 0.0726 | 0.8135 | **1.311** | 0.1593 |
| MapTRv2-Centerline (Liao et al., 2023b) + Ours | **0.3427** (−8%) | **0.6763** (−10%) | **0.0570** (−22%) | **0.7419** (−9%) | 1.341 (+2%) | **0.1506** (−6%) |
| StreamMapNet (Yuan et al., 2024) | 0.3972 | 0.8186 | 0.0926 | 0.9492 | 1.740 | 0.2569 |
| StreamMapNet (Yuan et al., 2024) + (Gu et al., 2024) | 0.3848 | 0.7954 | 0.0861 | 0.9036 | 1.645 | 0.2359 |
| StreamMapNet (Yuan et al., 2024) + Ours | **0.3711** (−7%) | **0.7745** (−10%) | **0.0796** (−22%) | **0.8065** (−11%) | **1.600** (−3%) | **0.2418** (+2%) |

## 4 EXPERIMENT

### 4.1 EXPERIMENT SETUP

**Dataset.** We evaluate our method on the large-scale nuScenes (Caesar et al., 2020) dataset, which consists of 1,000 driving scenes, split into 500, 200, and 150 scenes for training, validation, and testing, respectively. Each scene spans approximately 20 seconds, with RGB images from six cameras covering a 360° horizontal field of view around the ego-vehicle. The sensor data is recorded at 10 Hz, and keyframe annotations are provided at 2 Hz. The dataset includes ground-truth (GT) HD maps, multi-sensor data, and tracked agent trajectories. Our work uses the same unified trajdata (Ivanovic et al., 2023) interface as in Gu et al. (2024) to standardize the transmission and conversion between the vectorized map estimation models and downstream prediction models. To ensure compatibility across various prediction and mapping models, we also leverage the method in Gu et al. (2024) of trajdata temporal interpolation utility (Ivanovic et al., 2023) to upsample the nuScenes trajectory data frequency from 2 Hz to 10 Hz, ensuring frequency alignment. Finally, each prediction model is tasked with predicting the future vehicle motion 3 seconds ahead, based on observations from the previous 2 seconds of vehicle movement.

**Metrics.** For evaluating trajectory prediction performance, we adopt four standard evaluation metrics that are commonly used in recent prediction challenges: minimum Average Displacement Error (minADE), minimum Final Displacement Error (minFDE), and Miss Rate (MR). For each agent model predicting six trajectories, the minADE metric evaluates the average Euclidean distance, in meters, between the most accurate predicted trajectory and the ground truth trajectory within the prediction range. The minFDE metric measures the error between the final predicted position of the trajectory and the ground truth. The best predicted trajectory is defined as the one with the smallest endpoint error. The MR metric refers to the proportion of the best predicted trajectory endpoints that exceed 2 meters compared to the ground truth trajectory endpoints.

### 4.2 QUANTITATIVE EVALUATION

To evaluate the effect of the proposed uncertainty on downstream vehicle trajectory prediction, we conduct experiments and comparisons with previous uncertainty methods on the six model combinations. The combinations are formed by combining the map information obtained from 3 existing high-definition map estimation methods (Liao et al., 2023a;b) with 2 downstream trajectory prediction methods (Zhou et al., 2022; Gu et al., 2021). **From the trajectory prediction aspects, we observe a consistent improvement in Table 1.** (1) Integration with MapTR: When using MapTR for map estimation with our positional and semantic uncertainty, the DenseTNT trajectory prediction method shows the most significant gains, with minADE, minFDE, and MR improving by 13%, 5%, and 2%, respectively. (2) Integration with MapTRv2: Although MapTRv2 outperforms MapTR in high-definition map estimation, its application to downstream trajectory prediction does not yield

Table 2: Ablation study on our main components, *i.e.*, positional uncertainty and semantic uncertainty. Unc_pos denotes the positional uncertainty method, while Unc_sem represents the semantic uncertainty method. We use a checkmark ✓ to indicate whether the method is applied. * means part of our uncertainty.

| Method | Unc_pos | Unc_sem | minADE ↓ | minFDE ↓ | MR ↓ |
|---|---|---|---|---|---|
| MapTR (Liao et al., 2023a) + HiVT (Zhou et al., 2022) | | | 0.4015 | 0.8418 | 0.0981 |
| Gu (Gu et al., 2024) + HiVT (Zhou et al., 2022) | | | 0.3854 | 0.7909 | 0.0834 |
| Ours* + HiVT (Zhou et al., 2022) | ✓ | | 0.3717 | 0.7820 | 0.0829 |
| Ours* + HiVT (Zhou et al., 2022) | | ✓ | 0.3643 | 0.7573 | 0.0812 |
| Ours + HiVT (Zhou et al., 2022) | ✓ | ✓ | **0.3660** | **0.7564** | **0.0745** |
| MapTR (Liao et al., 2023a) + DenseTNT (Gu et al., 2021) | | | 1.091 | 2.058 | 0.3543 |
| Gu (Gu et al., 2024) + DenseTNT (Gu et al., 2021) | | | 1.089 | 2.006 | 0.3499 |
| Ours* + DenseTNT (Gu et al., 2021) | ✓ | | 1.093 | 2.2067 | 0.4286 |
| Ours* + DenseTNT (Gu et al., 2021) | | ✓ | 0.9867 | 1.9346 | 0.3456 |
| Ours + DenseTNT (Gu et al., 2021) | ✓ | ✓ | **0.954** | **1.909** | **0.3429** |

a noticeable improvement and sometimes even leads to a decline. Incorporating our positional and semantic uncertainty, the performance improvement with MapTRv2-generated maps is comparable to that of MapTR. (3) Integration with MapTRv2-Centerline: Using MapTRv2-centerline, which includes lane centerlines in map estimation, and applying our uncertainties, both trajectory prediction methods achieve the best performance. For HiVT, minADE, minFDE, and MR improve by 8%, 10%, and 22%, respectively, compared to the baseline. The improvement for DenseTNT is less pronounced, but we still increase 9% miniADE. (4) Integration with StreamMapNet: As for using StreamMapNet for map construction with our positional and semantic uncertainty, both trajectory prediction methods also achieve the best performance, especially in the HiVT method, the MR metric improves by 22% than Gu et al. (2024). **From the map aspects, the HiVT trajectory prediction model shows greater improvements.** After applying our positional uncertainty and semantic uncertainty to all map methods, the improvement in MR is the most significant in HiVT, achieved an improvement of up to 22%, indicating that by incorporating our proposed map uncertainty, the prediction model can effectively adjust its behavior to better match the actual trajectory. Additionally, in DenseTNT, the most significant improvement resulting from the four map estimations using our uncertainty methods is reflected in the minADE metric, which achieved an improvement of up to 13%, showing that our uncertainty approach helps the model reduce extreme displacement situations and makes trajectory predictions more accurate. Overall, as shown in Table 1, the predicted maps obtained using our positional uncertainty and semantic uncertainty achieve a significant performance improvement in downstream vehicle trajectory prediction compared to the baseline and Gu et al. (2024) method.

Additionally, we also consider other prediction methods. We have basically reproduced (Deo & Trivedi, 2018; Mao et al., 2023; Wang et al., 2023) and applied its core idea to make trajectory prediction. The table 3 shows the trajectory prediction results combined with the original Maptrv2 online HD map (Maptrv2) and the Maptrv2 online HD map with our proposed uncertainty (Maptrv2 + Our uncertainty). It is observed that the performance of our uncertainty method still exceeds that without using uncertainty map information, indicating the generalization ability of our method in different prediction methods.

## 4.3 ABLATION STUDIES AND FURTHER DISCUSSION

In Table 2, we discuss the impact of our proposed positional uncertainty and semantic uncertainty on the two downstream trajectory prediction tasks. The table presents the results obtained by applying these two types of uncertainty to one of the map estimation methods, MapTR, and then using the resulting maps for trajectory prediction. The base method serves as a comparison, using the baseline from (Gu et al., 2024).

**Effectiveness of Positional Uncertainty.** We first compare the effect of introducing only positional uncertainty of map elements against the baseline method. On HiVT, all three trajectory prediction evaluation metrics show a decline, with minADE increasing the most by 7%. This indicates that the HiVT-based method is more sensitive to the accuracy of the positional information of map elements.

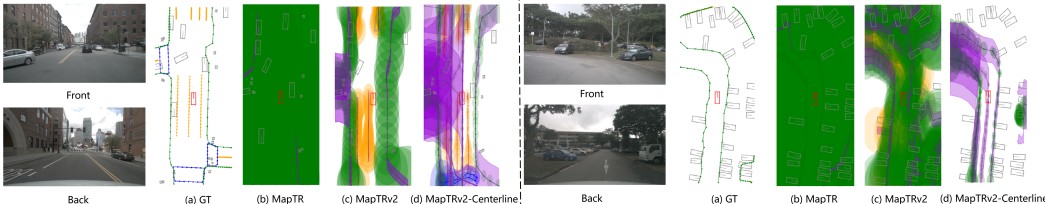

Figure 3: The left figure shows the effectiveness of our proposed method for estimating high-definition map positional uncertainty and semantic uncertainty in a normal road scenario in test set. The right figure also shows the effectiveness of our proposed method for estimating high-definition map positional uncertainty and semantic uncertainty in test set scenarios involving curved roads and parking lots. In the figure, green represents road boundaries, blue represents pedestrian crossings, orange represents lane dividers, purple indicates category semantic uncertainty, gray represents lane centerlines, the red vehicle denotes the ego vehicle, and the gray vehicles denote other agents.

In contrast, for DenseTNT, introducing only positional uncertainty to enhance map elements does not yield a significant improvement in trajectory prediction, and even leads to a decline in MR. This suggests that DenseTNT, utilizing GNN, is already capable of effectively leveraging positional relationships of map elements.

**Effectiveness of Semantic Uncertainty.** When introducing only semantic uncertainty to enhance map elements, the performance on HiVT declines more significantly compared to using positional uncertainty alone, particularly with a 9% increase in minADE compared to the baseline. For DenseTNT, the introduction of semantic uncertainty yields substantial improvements, with minADE increasing by 10% and minFDE by 7%. This demonstrates that the accuracy of semantic information plays a crucial role in enhancing trajectory prediction in complex and occluded scenarios. Since there are inherent errors in map estimation compared to ground truth, incorporating uncertainty in category information can better assist the trajectory prediction model.

Overall, applying both positional and semantic uncertainty map information to the HiVT model results in more noticeable improvements compared to DenseTNT. Introducing positional and semantic uncertainty information into the HiVT trajectory prediction model consistently enhances predictions, with semantic uncertainty showing a greater impact. Notably, when both uncertainties are utilized together, the MR metric for HiVT decreases significantly, whereas the decline is minimal when using either one individually. This highlights that the proposed positional and semantic uncertainties are indispensable and complementary to each other.

**Map Uncertainty Visualization.** Figure 3 illustrate the visualization effects of the two uncertainties we introduced across the three map estimation methods. The top of the figure shows a scenario where tall buildings on both sides of the road obscure the intersection, and the presence of other vehicles and pedestrians results in incomplete information captured by the camera of the vehicle, leading to high uncertainty in the map model prediction. It can be observed that the MapTR model generates high levels of positional and semantic uncertainty, whereas MapTRv2 and MapTRv2-centerline exhibit lower uncertainty. However, the obscured intersection causes these models to produce higher positional and semantic uncertainty at the road junction. The bottom of the figure illustrates a parking lot environment, where road boundaries are unclear and there are no distinct driving lanes, with many surrounding vehicles obscuring the road conditions. Here, our positional and semantic uncertainties are particularly evident at the turns, reflecting the changes in the road under such conditions.

**Uncertainty-aware Trajectory Prediction Visualization.** To better illustrate the improvement in map trajectory prediction brought by our proposed positional uncertainty and semantic uncertainty, we visualize the enhancement effects in some typical scenarios using our two types of uncertainty in Figure 4. For a clearer representation of how these uncertainties supplement map information, we choose MapTRv2 to generate visualization images with two trajectory prediction models, since the uncertainties generated by MapTR is more significant and less visually intuitive on the map.
**(1) Complex Urban Intersections.** As shown in Figure 4 top left, we evaluate vehicle trajectory predictions at a complex intersection with additional turning lanes. The figure includes the ground truth (GT) of the map and vehicle trajectories. We observe that using HiVT and DenseTNT as inputs for the downstream trajectory prediction tasks, with the same map uncertainty, results in good

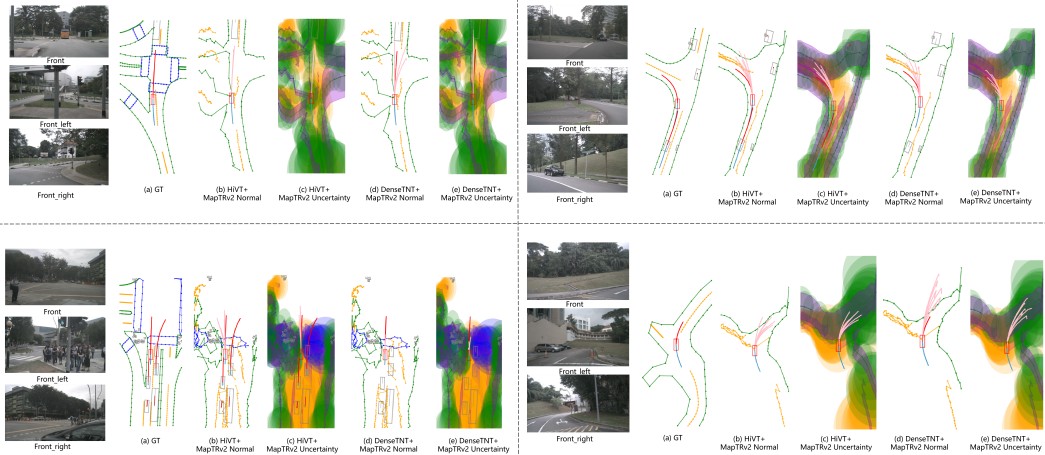

Figure 4: **Top Left:** At busy and complex intersections, where map elements are dense, our proposed uncertainty information enhances vehicle trajectory prediction. **Top Right:** When the vehicle is turning, the camera perspective may not fully capture all the surrounding road conditions, which could lead to trajectory predictions extending beyond the road boundaries. **Bottom Left:** When the surrounding environment is complex with numerous occlusions, both types of uncertainties in map prediction increase. **Bottom Right:** When lane information in the map environment is unclear, the surroundings are open, and the model map estimation is poor, our uncertainty information helps maintain high accuracy in vehicle trajectory prediction even with incomplete map information input.

prediction performance, reducing routing errors and effectively handling such multi-lane scenarios. Especially in the case of DenseTNT, the vehicle's predicted trajectory is noticeably closer to the ground truth due to the additional support from both types of map uncertainty. **(2) Vehicle Turning Scenario.** In Figure 4 top right, we show a scenario where the vehicle is about to turn, and the trajectory prediction model is prone to large errors due to unclear road boundaries and camera perspective issues. By incorporating our two types of uncertainty in the map information, it can be clearly seen that the vehicle trajectory in both methods is more reasonable, avoiding situations where the trajectory exceeds road boundaries when no uncertainty is introduced. **(3) Traffic Situation with Significant Occlusion.** As shown in Figure 4 bottom left, we present the improvement in model trajectory prediction in a complex traffic situation with significant occlusion and many pedestrians. When many pedestrians obscure the road information, our introduced uncertainties are reflected in darker colors, indicating the model uncertainty about both the positional and semantic information in these areas. Without such uncertainty assistance, the model predicted trajectory can be seen to deviate significantly, suggesting that the vehicle would drive toward the pedestrians. By incorporating both positional and semantic uncertainty, the model considers these uncertainties and predicts a more reasonable trajectory. **(4) Unclear Map Environment.** Figure 4 bottom right illustrates a scenario where the road environment is relatively open, the road information is vague, and the existing map estimation models are unable to accurately estimate all map elements. By introducing the two types of uncertainty—positional and semantic uncertainty—we can supplement the map information, resulting in more accurate model predictions. The figure shows that when there are fewer map elements without uncertainty supplementation, the vehicle trajectory tends to drift beyond the road boundaries. However, after introducing the uncertainty information, the situation is alleviated, allowing for reasonable trajectory prediction despite the lack of complete map element information.

As shown in Figure 5, (a) depicts a rainy scene. It includes visual distortions like the cloudy weather, the reflection of the surrounding environment due to the water on the road surface, and the raindrops. These distortions will obscure the camera's ability and introduce potential uncertainty to complicate the identification of road edges and lane markings. Through our uncertainty estimation, we can effectively identify and quantify the uncertainty in road detection under such weather conditions, enabling the vehicle to maintain the correct path. As seen from this Figure, the circle of position and semantic uncertainty of the map is larger and deeper in places where the line of sight is obscured or blurred, such as when the sidewalk is obscured by vehicles and rain. Our method has performed

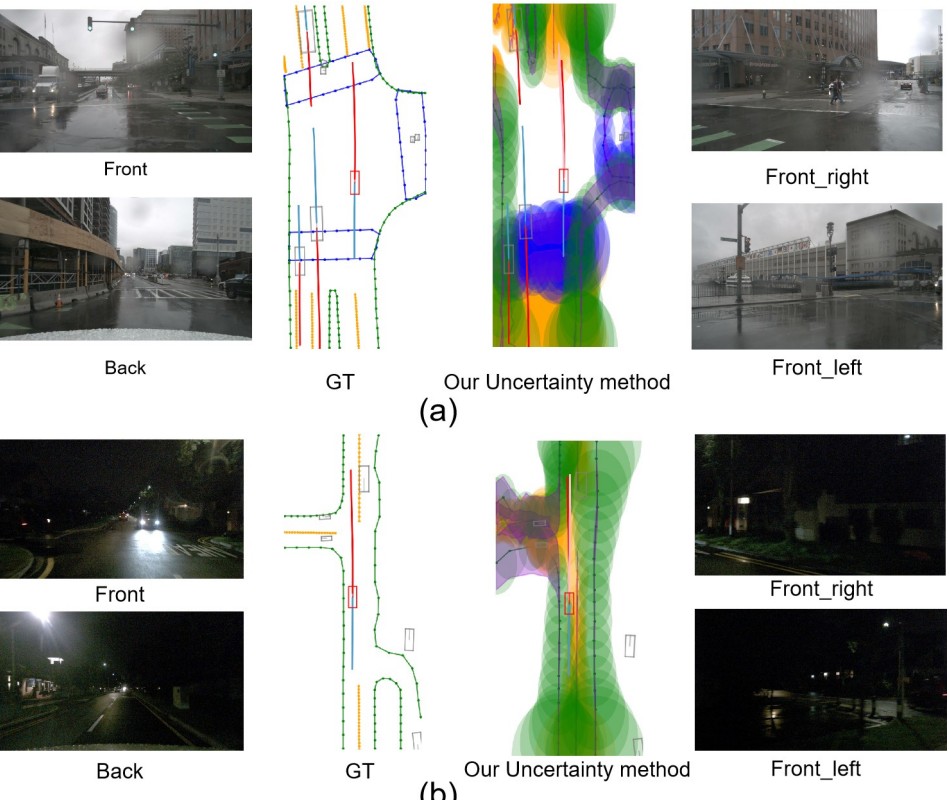

Figure 5: **(a)** Rainy day scene. **(b)** The night scene.

Table 3: Three trajectory prediction results with one online HD map on the nuScenes dataset. "Maptrv2": Original Maptrv2 online HD map (Liao et al., 2023b), "Maptrv2 + our uncertainty": Maptrv2 online HD map with our proposed uncertainty.

| Online HD Map Method | CSP (Deo & Trivedi, 2018) | | | Wsip (Wang et al., 2023) | | | Leapfrog (Mao et al., 2023) | | |
|---|---|---|---|---|---|---|---|---|---|
| | minADE ↓ | minFDE ↓ | MR ↓ | minADE ↓ | minFDE ↓ | MR ↓ | minADE ↓ | minFDE ↓ | MR ↓ |
| Maptrv2 | 0.9037 | 1.733 | 0.2876 | 0.3752 | 0.7837 | 0.0849 | 1.0392 | 1.8995 | 0.3013 |
| Maptrv2 + our uncertainty | 0.8630 | 1.639 | 0.2737 | 0.3736 | 0.7871 | 0.0803 | 0.9627 | 1.7749 | 0.2589 |

a good estimation and construction. (b) shows a night scene. Poor visibility makes it difficult for image sensors to capture accurate road information, so map estimation produces higher uncertainty regarding road locations compared to daytime scenes. Notably, at an obscured intersection hidden behind trees on the left side of the vehicle's path, our method effectively highlights semantic uncertainty (indicated in purple) and positional uncertainty in the road and lane lines. Additionally, the positional uncertainty is significantly higher at the end of the field of view, aligning well with the expected judgment for real-world vehicle navigation.

Figure 6 shows the HD map visualization on Argoversev2 sensor dataset after applying our uncertainty method. (a) In the case of a normal driving road, the semantic uncertainty and location uncertainty of the road boundary obscured by trees at the rear corner are the largest, while the semantic uncertainty of other road information is very small. (b) and (c) reflect the uncertainty of our map under complex traffic intersections. We can see that the semantic uncertainty and positional uncertainty are both large at the intersection, especially at the left and right corners, because the line of sight is obscured. At the same time, the presence of turning vehicles at the intersection, such as the oil tanker in (c), leads to the occlusion of the road boundary and pedestrian line, which is reflected in our estimated map that the two kinds of uncertainties are very large (darker and wider). When the road boundary on both sides is clear and unobstructed, the semantic uncertainty is very small, and the location uncertainty is also smaller than that of the intersection that is unobstructed.

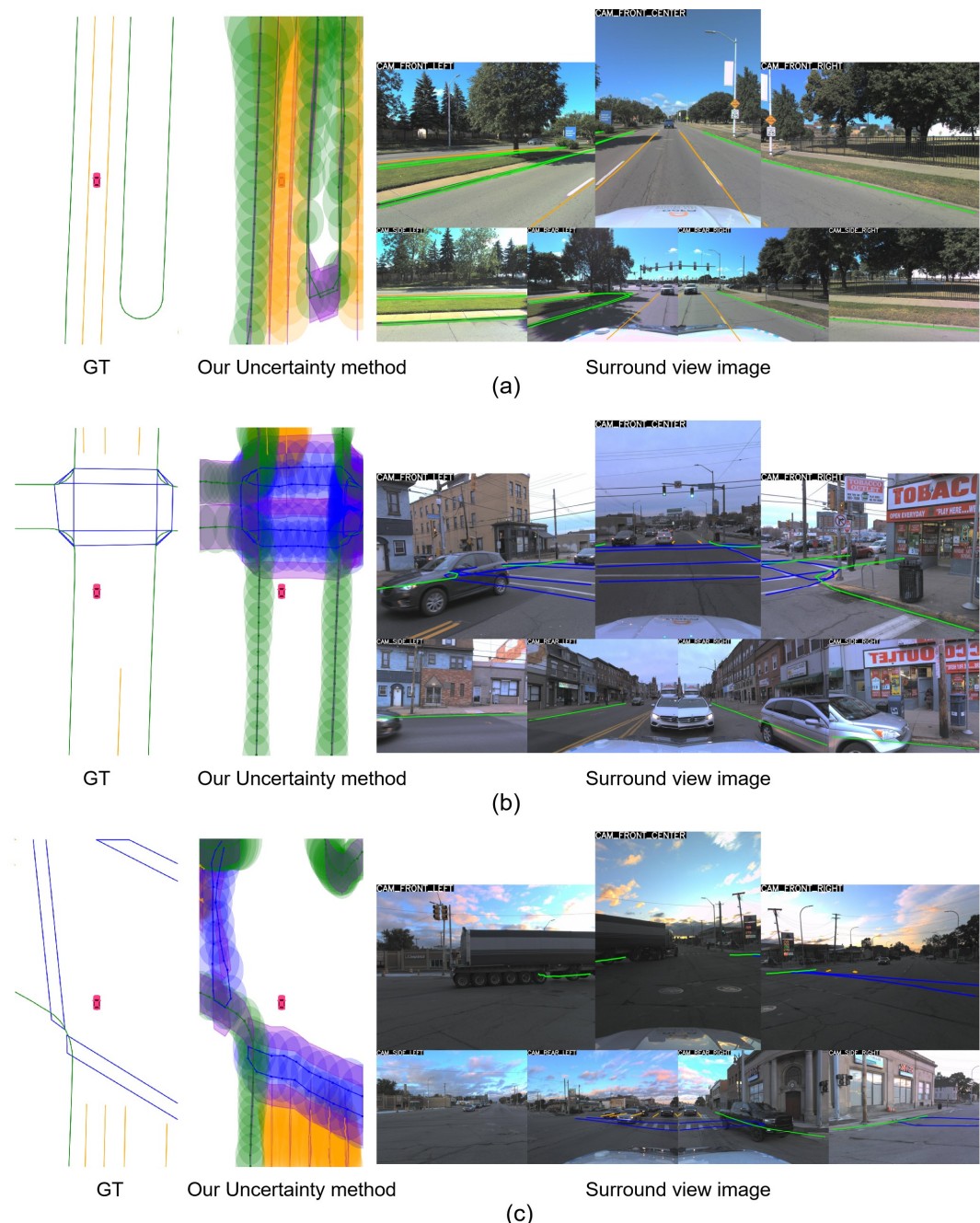

Figure 6: Map visualization of our uncertainty method in the Argoversev2 sensor dataset. In the figure, green represents road boundaries, blue represents pedestrian crossings, orange represents lane dividers, purple indicates category semantic uncertainty, the red vehicle denotes the ego vehicle.

## 5 CONCLUSION

In this work, we propose a general vectorized high-definition map uncertainty estimation method to solve map data noise for downstream vehicle trajectory prediction tasks, incorporating an auxiliary head to regress positional and semantic uncertainties. We enhance several state-of-the-art online map estimation methods, including MapTR (Liao et al., 2023a), MapTRv2 (Liao et al., 2023b), MapTRv2-Centerline (Liao et al., 2023b) and StreamMapNet (Yuan et al., 2024), with our method that incorporates both types of uncertainties to produce map elements with positional and semantic

uncertainty information. The resulting uncertainty-enhanced map elements are then fed into state-of-the-art trajectory prediction methods DenseTNT (Gu et al., 2021) and HiVT (Zhou et al., 2022). The results show that our proposed uncertainty-enhanced map elements significantly improve the performance of the prediction models, with maximum improvements of 8%, 10%, and 22% in minADE, minFDE, and MR, respectively.

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

## A  IMPLEMENTATION DETAILS

All models are trained using four NVIDIA GeForce RTX A6000 GPUs, each with 49 GB of memory. we employ four independent methods and adjust the network structures to account for positional and semantic uncertainty, resulting in slight model parameter changes compared to the baseline. Additionally, since the models differ in structure, we apply separate hyperparameter settings for each, as shown in Table 4. For all four map estimation model, we set learning rate to 1.0E-4, regression loss weigh to 0.03 and gradient norm to 3. Other training detials are following the base model.

Similarly, for the two downstream trajectory prediction models, the model input information changes and the model structures differ. Therefore, we use different hyperparameters for training each model, as referenced in Table 5. We set learning rate to 3.5E-4 for all four map prediction model with trajectory prediction model HiVT. Four different learning rates form 2.5E-3 to 3.5E-3 are set for different map prediction models with trajectory prediction model DenseTNT. When use the HiVT model, the batch size is set to 32, as for DenseTNT, batch size set to 16. The dropout rate for all trajectory prediction models are 0.1. All other hyperparameters in these two trajectory prediction models are unchanged.

| Method | Regression Loss Weight | LR | Gradient Norm |
|---|---|---|---|
| MapTR (Liao et al., 2023a) | 0.03 | 1.0E-4 | 3 |
| MapTRv2 (Liao et al., 2023b) | 0.03 | 1.0E-4 | 3 |
| MapTRv2-Centerline (Liao et al., 2023b) | 0.03 | 1.0E-4 | 3 |
| StreamMapNet (Yuan et al., 2024) | 2 | 1.0E-4 | 3 |

Table 4: Map prediction training hyperparameters.

| Online HD Map Method | LR | Batch Size | Dropout |
|---|---|---|---|
| MapTR (Liao et al., 2023a) + HiVT (Zhou et al., 2022) | 3.5E-4 | 32 | 0.1 |
| MapTR (Liao et al., 2023a) + DenseTNT (Gu et al., 2021) | 3.0E-3 | 16 | 0.1 |
| MapTRv2 (Liao et al., 2023b) + HiVT (Zhou et al., 2022) | 3.5E-4 | 32 | 0.1 |
| MapTRv2 (Liao et al., 2023b) + DenseTNT (Gu et al., 2021) | 2.0E-3 | 16 | 0.1 |
| MapTRv2-Centerline (Liao et al., 2023b) + HiVT (Zhou et al., 2022) | 3.5E-4 | 32 | 0.1 |
| MapTRv2-Centerline (Liao et al., 2023b) + DenseTNT (Gu et al., 2021) | 3.5E-3 | 16 | 0.1 |
| StreamMapNet (Yuan et al., 2024) + HiVT (Zhou et al., 2022) | 3.5E-4 | 32 | 0.1 |
| StreamMapNet (Yuan et al., 2024) + DenseTNT (Gu et al., 2021) | 1E-3 | 16 | 0.1 |

Table 5: Hyperparameters chosen for different trajectory prediction methods.

# B  MATHEMATICAL PROOF OF USING PREDICTION DIFFERENCES BETWEEN MAIN AND AUXILIARY CLASSIFIERS AS A MEASURE OF MODEL UNCERTAINTY

## B.1  DEFINITIONS AND ASSUMPTIONS

1. **Model Structure:** The deep learning model $M$ includes a main head $C_{\text{main}}$ and an auxiliary head $C_{\text{aux}}$. For an input sample $x$, the prediction output of the main head is $p_{\text{main}} = C_{\text{main}}(x)$, and the prediction output of the auxiliary head is $p_{\text{aux}} = C_{\text{aux}}(x)$.

2. **Uncertainty:** We focus on the model's *Epistemic Uncertainty*, which is the uncertainty in the model parameters. Assume the model parameters $\theta$ are random variables with a prior distribution $P(\theta)$.

3. **Prediction Difference:** Define the prediction difference $D(x)$ as:

$$D(x) = \|p_{\text{main}} - p_{\text{aux}}\|,$$

where $\|\cdot\|$ denotes a norm (*e.g.*, L2 norm).

## B.2  MATHEMATICAL DERIVATION

**Model's Predictive Distribution.** Assume the model's output is a probability distribution $P(y|x, \theta)$, where $y$ is the class label, $x$ is the input sample, and $\theta$ are the model parameters.

**Posterior Predictive Distribution.** According to Bayes' theorem, the posterior predictive distribution of the model can be expressed as:

$$P(y|x) = \int P(y|x, \theta)P(\theta|x)d\theta,$$

where $P(\theta|x)$ is the posterior distribution of the model parameters.

**Parameter Uncertainty.** The uncertainty in the parameters can be measured by the variance of the posterior distribution:

$$\text{Var}(\theta|x) = \mathbb{E}_{\theta|x}[(\theta - \mathbb{E}_{\theta|x}[\theta])^2] = \mathbb{E}_{\theta|x}[\theta^2] - (\mathbb{E}_{\theta|x}[\theta])^2.$$

**Prediction Difference and Parameter Uncertainty.** To relate the prediction difference $D(x)$ to parameter uncertainty, we need to consider the predictions of the main and auxiliary heads. Assume the parameters of the main head and auxiliary head are $\theta_{\text{main}}$ and $\theta_{\text{aux}}$, respectively, and they have the same prior distribution, *i.e.*, $P(\theta_{\text{main}}) = P(\theta_{\text{aux}})$.

The predictions of the main and auxiliary heads can be expressed as:

$$p_{\text{main}} = \mathbb{E}_{\theta_{\text{main}}|x}[P(y|x, \theta_{\text{main}})].$$

$$p_{\text{aux}} = \mathbb{E}_{\theta_{\text{aux}}|x}[P(y|x, \theta_{\text{aux}})].$$

**Expression for Prediction Difference.** Assume the difference in predictions can be approximated by First-order Taylor Expansion:

$$p_{\text{main}} - p_{\text{aux}} \approx \mathbb{E}_{\theta|x}[\nabla_\theta P(y|x, \theta) \cdot (\theta_{\text{main}} - \theta_{\text{aux}})]$$

where $\nabla_\theta P(y|x, \theta)$ is the gradient of $P(y|x, \theta)$ with respect to $\theta$.

Thus, the prediction difference $D(x)$ can be expressed as:

$$D(x) = \|p_{\text{main}} - p_{\text{aux}}\| \approx \|\mathbb{E}_{\theta|x}[\nabla_\theta P(y|x, \theta) \cdot (\theta_{\text{main}} - \theta_{\text{aux}})]\|$$

**Relationship Between Prediction Difference and Parameter Uncertainty**

To simplify the analysis, assume $\theta_{\text{main}}$ and $\theta_{\text{aux}}$ are independently and identically distributed (i.i.d.). Then:

$$\mathbb{E}_{\theta|x}[\|\nabla_\theta P(y|x, \theta) \cdot (\theta_{\text{main}} - \theta_{\text{aux}})\|^2] \approx \mathbb{E}_{\theta|x}[\|\nabla_\theta P(y|x, \theta)\|^2] \cdot \mathbb{E}_{\theta|x}[(\theta_{\text{main}} - \theta_{\text{aux}})^2]$$

Noting that $\mathbb{E}_{\theta|x}[(\theta_{\text{main}} - \theta_{\text{aux}})^2] = 2(\mathbb{E}_{\theta|x}[\theta^2] - (\mathbb{E}_{\theta|x}[\theta])^2) = 2 \cdot \text{Var}(\theta|x)$, we have:

$$\mathbb{E}_{\theta|x}[\|\nabla_\theta P(y|x, \theta) \cdot (\theta_{\text{main}} - \theta_{\text{aux}})\|^2] \approx 2 \cdot \mathbb{E}_{\theta|x}[\|\nabla_\theta P(y|x, \theta)\|^2] \cdot \text{Var}(\theta|x)$$

Assuming $k = \mathbb{E}_{\theta|x}[\|\nabla_\theta P(y|x, \theta)\|^2]$, which is positive, we get:

$$\mathbb{E}_{\theta|x}[\|\nabla_\theta P(y|x, \theta) \cdot (\theta_{\text{main}} - \theta_{\text{aux}})\|^2] \approx 2k \cdot \text{Var}(\theta|x)$$

Thus, the prediction difference $D(x)$ can be expressed as:

$$D(x) \approx \sqrt{2k \cdot \text{Var}(\theta|x)}$$

Simplifying further, we obtain:

$$D(x) \propto \sqrt{\text{Var}(\theta|x)}$$

### B.3 CONCLUSION

From the above derivation, we have shown that the prediction difference $D(x)$ is proportional to the square root of the model's uncertainty $\sqrt{\text{Var}(\theta|x)}$. Therefore, the prediction difference $D(x)$ can serve as a measure of the model's uncertainty for a given sample.

> The prediction difference $D(x)$ is proportional to the square root of the model's uncertainty $\sqrt{\text{Var}(\theta|x)}$.

### B.4 COMPARISON BETWEEN GU ET AL. AND OURS

**1. Gu et al. (2024) using Class Probability as Uncertainty**

Definition of class probability $P(y|x, \theta)$: The probability that input $x$ belongs to class $y$ given model parameters $\theta$.

Uncertainty measure: The uncertainty is quantified directly using $P(y|x, \theta)$ itself.

**2. Ours using Prediction Difference Between Two Heads.**

Two independent heads: The main head with parameters $\theta_{\text{main}}$ and the auxiliary head with parameters $\theta_{\text{aux}}$.

Uncertainty measure: The difference $D(x) = \|p_{\text{main}} - p_{\text{aux}}\|$ quantifies uncertainty due to variability in $\theta$.

**3. Comparison.**

(1) For a given $\theta$, $P(y|x,\theta)$ is fixed and does not account for uncertainty in $\theta$.

(2) Our prediction difference $D(x)$:

$$D(x) = \|p_{\text{main}} - p_{\text{aux}}\| \propto \sqrt{\text{Var}(\theta|x)}.$$

Thus, $D(x)$ captures how sensitive the predictions are to changes in $\theta$ and is directly related to parameter uncertainty $\text{Var}(\theta|x)$. Larger variance in parameters leads to larger $D(x)$, indicating higher uncertainty.

