# OpenReview forum: "Leveraging Semantic and Positional Uncertainty for Trajectory Prediction"
_ICLR.cc/2025/Conference — Submitted to ICLR 2025_

### Official Review · Reviewer_Zri7 · 2024-10-30

**Soundness:** 2
**Presentation:** 3
**Contribution:** 2
**Rating:** 3
**Confidence:** 4

**Summary:**

The paper studies an important problem of trajectory prediction, which considers both position and semantic uncertainties. The authors propose a dual-head structure to model the two uncertainties explicitly. Experiments show the effectiveness of the proposed method to some extent.

**Strengths:**

1. The paper is well-written and easy to follow.
2. The proposed method considers both positional and semantic uncertainties.
3. Experiments show the effectiveness of the proposed method.

**Weaknesses:**

1. Although the proposed method studies the positional and semantic uncertainties explicitly, the corresponding uncertainty modeling seems to be very simple and lacks enough technical contribution. It would be better to provide more details to show the novelty and technical contributions of this paper.
2. It is not enough to show the effectiveness of the proposed method with only one dataset. It is encouraged to include more datasets, for example, NGSIM [1] and highD [2].

[1]. Convolutional social pooling for vehicle trajectory prediction, CVPR 2018.

[2]. The highD dataset: A drone dataset of naturalistic vehicle trajectories on german highways for validation of highly automated driving systems, ITSC 2018.

3. It is suggested to include more prediction methods [3, 4, 5] in Table 1, which could be more comprehensive to show the effectiveness of the proposed method.

[3]. Leapfrog diffusion model for stochastic trajectory prediction, CVPR 2023.

[4]. Wsip: Wave superposition inspired pooling for dynamic interactions-aware trajectory prediction, AAAI 2023.

[5]. Convolutional social pooling for vehicle trajectory prediction, CVPR 2018.

4. It would be better to provide a case study to intuitively show the model can learn the uncertainties.

**Questions:**

1. What is the technical contribution of the proposed method? Uncertainty modeling seems to be very simple.
2. Why does not include more datasets and baselines?

---

> ### Author Response · Authors · 2024-11-23
> **Response to Reviewer Zri7 (Part 1)**
>
> We sincerely thank Reviewer Zri7 for the valuable feedback and thoughtful comments on our paper. We are also pleased to hear your positive recognition that the paper is well-written and easy to follow. In the following, we will provide detailed responses to each of your questions and weaknesses. We believe that your suggestions will significantly improve the coherence, quality, and overall impact of our paper.
>
> **Q1:** Uncertainty modeling seems to be very simple and lacks enough technical contribution.
> Thank you for your question. We provide more details to show the novelty and technical contributions of our uncertainty-based method here.
>
> (1) **Novelty:** (i) Uncertainty is quite important for safety, but remains underexplored in trajectory predictions.
>
> (ii) We explore two kinds of uncertainty, i.e., positional and semantic uncertainty.
>
> (iii) Albeit simple, the proposed method achieves competitive results, and is compatible with various mapping construction and trajectory prediction algorithms.
>
> (2) **Technical Mechanism:**
> (i) Two-head Mechanism: We use two classifiers to obtain \mu and \mu{'} for uncertainty estimation. The idea of two classifiers is also common in segmentation, depth, and other tasks [2, 3]. It, de facto, is similar to Bayesian network with two different weights. Thereby, the variance also indicates the uncertainty. More discussion can be found in "Why use an auxiliary head to estimate uncertainty?" (Line 262-272)
>
> (ii) Dropout Mechanism: The usage of dropout is a widely adopted method for uncertainty estimation. Dropout serves as an approximation to Bayesian neural networks [1], effectively transforming a standard neural network into a Bayesian network. Bayesian networks are a common and stable approach for uncertainty estimation. As demonstrated by [1], dropout can reliably estimate uncertainty. Therefore, we have employed dropout layers in our model to leverage these benefits.
>
> (3) **Performance:** As seen in Table 1, our method performs better by surpassing Gu et al. 5% minADE in Maptr online map construction method with HiVT trajectory prediction method and 13% minADE in DenseTNT trajectory prediction method.
>
> [1] Dropout as a Bayesian Approximation: Representing Model Uncertainty in Deep Learning.
>
> [2] Rectifying Pseudo Label Learning via Uncertainty Estimation for Domain Adaptive Semantic Segmentation.
>
> [3] Harnessing Uncertainty-aware Bounding Boxes for Unsupervised 3D Object Detection.
>
>
> **Q2:** Encouraged to include more datasets, for example, NGSIM and highD.
>
> Following the setting of the previously published work [7,9], we try to combine the vectorized HD maps with uncertainty and improve trajectory prediction. That means HD maps and trajectory annotations for the same scene are required. For now, the most commonly available datasets do not satisfy such a requirement. They either have HD maps or trajectory annotation, or some datasets provide both but for different scenes. nuScene is the dataset that can satisfy this requirement. The datasets such as NGSIM, HighD, InD [10,11,12], contain only the trajectory data and thus can not be used by our method. Besides, most existing works in autonomous driving adopt only one dataset for evaluation [4,5,6,7,8,9].
>
> Nonetheless, we have searched online for other potential datasets. We are now experimenting with the Argoverse v2 sensor dataset and will report the results immediately once we get the results. If you have other datasets recommended, we would love to try and run on them.
>
> [4] St-p3: End-to-end vision-based autonomous driving via spatial-temporal feature learning (ECCV 2022)
>
> [5] UniAD: Planning-oriented Autonomous Driving (CVPR 2023 Best Paper)
>
> [6] VAD: Vectorized Scene Representation for Efficient Autonomous Driving (ICCV 2023)
>
> [7] Producing and Leveraging Online Map Uncertainty in Trajectory Prediction (CVPR 2024)
>
> [8] GenAD: Generative End-to-End Autonomous Driving (ECCV 2024)
>
> [9] Accelerating Online Mapping and Behavior Prediction via Direct BEV Feature Attention (ECCV 2024)
>
> [10] Convolutional social pooling for vehicle trajectory prediction (CVPR 2018)
>
> [11] The highD dataset: A drone dataset of naturalistic vehicle trajectories on german highways for validation of highly automated driving systems(ITSC 2018)
>
> [12] The inD Dataset: A Drone Dataset of Naturalistic Road User Trajectories at German Intersections

---

> ### Author Response · Authors · 2024-11-23
> **Response to Reviewer Zri7 (Part 2)**
>
> **Q3:** Include more prediction methods.
>
> Thanks a lot for the great suggestion and recommendation. At present, we have basically reproduced [13] and applied its core idea to the convolutional social pooling layer to learn the interaction among vehicles. The following table shows the trajectory prediction results combined with the original Maptrv2 online HD map (Maptrv2) and the Maptrv2 online HD map with our proposed uncertainty (Maptrv2 + Our uncertainty). It is observed that the performance of our uncertainty method still exceeds that without using uncertainty map information, indicating the generalization ability of our method in different prediction methods. We add Table 3 in our paper (Line 572-579).
>
> | **Method**                  | **minADE ↓** | **minFDE ↓** | **MR ↓** |
> |-----------------------------|--------------|--------------|----------|
> | Maptrv2                     | 0.9037       | 1.733        | 0.2876   |
> | Maptrv2 + Our Uncertainty   | 0.8630       | 1.639        | 0.2737   |
>
> We are still reproducing the rest prediction methods and combining with our uncertainty map information to make trajectory predictions. We will post the results as soon as the experiments are finished.
>
> [13] Convolutional social pooling for vehicle trajectory prediction, CVPR 2018.
>
> **Q4:** Case study to intuitively show the model can learn uncertainties.
>
> Thank you. In the paper, we added Figure 5（See Line 540-570）to intuitively show the model can learn uncertainties. Figure 5 (a) depicts a rainy scene. It includes visual distortions like the cloudy weather, the reflection of the surrounding environment due to the water on the road surface, and the raindrops. These distortions will obscure the camera’s ability and introduce potential uncertainty to complicate the identification of road edges and lane markings. Through our uncertainty estimation, we can effectively identify and quantify the uncertainty in road detection under such weather conditions, enabling the vehicle to maintain the correct path. As seen from this Figure, the circle of position and semantic uncertainty of the map is larger and deeper in places where the line of sight is obscured or blurred, such as when the sidewalk is obscured by vehicles and rain. Our method has performed a good estimation and construction.
>
> Figure 5 (b) shows a night scene. Poor visibility makes it difficult for image sensors to capture accurate road information, so map estimation produces higher uncertainty regarding road locations compared to daytime scenes. Notably, at an obscured intersection hidden behind trees on the left side of the vehicle’s path, our method effectively highlights semantic uncertainty (indicated in purple) and positional uncertainty in the road and lane lines. Additionally, the positional uncertainty is significantly higher at the end of the field of view, aligning well with the expected judgment for real-world vehicle navigation.
>
> In conclusion, the cases above show the model obtained is able to capture the uncertainties and our proposed uncertainty method can effectively help vehicles predict trajectories under night and rainy conditions respectively.

---

> ### Comment · Reviewer_Zri7 · 2024-11-25
>
> Thanks for the response, which has addressed most of my concerns. However, I would like to see a comparison between the proposed method and the following SOTA studies [3, 4], which consider uncertainties either explicitly or implicitly. Thus, I would like to keep my rating.
>
> [3]. Leapfrog diffusion model for stochastic trajectory prediction, CVPR 2023.
>
> [4]. Wsip: Wave superposition inspired pooling for dynamic interactions-aware trajectory prediction, AAAI 2023.

---

> ### Author Response · Authors · 2024-11-28
> **Response to Reviewer Zri7**
>
> Thanks for your quick reply, we have completed the remaining two [3,4] experiments you mentioned. Based on the core ideas of these two methods, we basically reproduce them and apply them to our trajectory prediction task. the table below shows the complete results of the three new trajectory prediction tasks combined with the original Maptrv2 online HD map (Maptrv2) and the Maptrv2 online HD map with our proposed uncertainty (Maptrv2 + Our uncertainty). By comparing the results of these three trajectory prediction tasks,  it can be clearly seen that our uncertainty method still exceeds that without using uncertainty map information. We have updated Table 3 in pdf to include the full results of the three prediction methods. (Line 567-575).
>
> | Online HD Map Method         | CSP[5] minADE↓ | CSP[5] minFDE↓ | CSP[5] MR↓ | Wsip[4] minADE↓ | Wsip[4] minFDE↓ | Wsip[4] MR↓ | Leapfrog[3] minADE↓ | Leapfrog[3] minFDE↓ | Leapfrog[3] MR↓ |
> |------------------------------|--------------|--------------|----------|---------------|---------------|----------|-------------------|-------------------|---------------|
> | **Maptrv2**                  | 0.9037       | 1.733        | 0.2876   | 0.3752        | 0.7837        | 0.0849   | 1.0392            | 1.8995            | 0.3013        |
> | **Maptrv2 + our uncertainty** | 0.8630       | 1.639        | 0.2737   | 0.3736        | 0.7871        | 0.0803   | 0.9627            | 1.7749            | 0.2589        |
>
> [3]. Leapfrog diffusion model for stochastic trajectory prediction, CVPR 2023.
>
> [4]. Wsip: Wave superposition inspired pooling for dynamic interactions-aware trajectory prediction, AAAI 2023.
>
> [5]. Convolutional social pooling for vehicle trajectory prediction, CVPR 2018.

---

### Official Review · Reviewer_j4Ht · 2024-11-01

**Soundness:** 3
**Presentation:** 2
**Contribution:** 3
**Rating:** 6
**Confidence:** 4

**Summary:**

The manuscript identifies two types of uncertainties in trajectory prediction tasks: positional uncertainty and semantic uncertainty. It then proposes a framework to estimate these uncertainties and incorporates them into the trajectory prediction process in an end-to-end manner. The authors validate the effectiveness of the proposed framework on the nuScenes dataset using multiple existing trajectory prediction methods.

**Strengths:**

1. The motivation of the manuscript is valid. HD maps are crucial for motion prediction in autonomous driving scenarios, but estimated HD maps often contain noise and errors, introducing uncertainty into motion prediction systems. The authors recognize this issue and attempt to mitigate its impact, which benefits the community.

2. The proposed framework easily generalizes to various HD map generation methods as well as alternative motion prediction methods.

3. The experimental results and ablation studies are promising.

4. The manuscript is well-written. This reviewer particularly appreciates the discussion section, which makes the methodology easier to follow.

**Weaknesses:**

1. Testing on only one dataset may raise concerns about the framework's generalizability to more complex scenes.

2. More theoretical support may be needed for using an auxiliary head for uncertainty estimation, and the uncertainty visualization could be made easier to interpret.

**Questions:**

1. What are res5c and res4f in line 94.
2. How did author generate Figure 1 (b).

---

> ### Author Response · Authors · 2024-11-23
> **Response to Reviewer j4Ht (Part 1)**
>
> We sincerely appreciate the constructive feedback from Reviewer j4Ht. Thank you for the positive opinions regarding the paper motivation, the framework, and the results of experiments. The raised questions also encourage us to continue refining our approach. In the following section, we will answer the questions and concerns by providing detailed responses and clarifications.
>
> **Q1:** Concerns about the generalizability to more datasets and complex scenes.
>
> (1) Following the setting of the previously published work [4,6], we try to combine the vectorized HD maps with uncertainty and improve trajectory prediction. That means HD maps and trajectory annotations for the same scene are required. For now, most commonly available datasets do not satisfy such a requirement. They either have HD maps or trajectory annotation, or some datasets provide both but for different scenes. nuScene is the dataset that can satisfy this requirement. The datasets such as NGSIM, HighD, InD [7,8,9], contain only the trajectory data and thus can not be used by our method. Besides, most existing works in autonomous driving adopt only one dataset for evaluation [1,2,3,4,5,6].
>
> Nonetheless, we have searched online for other potential datasets. We are now experimenting with the Argoverse v2 sensor dataset and will report the results immediately once we get the results. If you have other datasets recommended, we would love to try and run on them.
>
> (2) Complex Scenarios: We visualize the normal road scenario and curved roads and parking lots scenarios in Figure 3 (Line 432-445). Furthermore, we add the visualization of a rainy and night scene in Figure 5 (Line 540-570) and some explanations (Line 533-539 and 581-587). We could observe that the proposed method could achieve robust uncertainty as well as trajectory prediction against challenging scenarios.
>
> **Q2:** More theoretical support may be needed and the uncertainty visualization could be made easier to interpret?
>
> Thank you for your insightful question. We provide more details of our method here.
>
> **(1) Two-head Mechanism:** We use two classifiers to obtain \mu and \mu{'} for uncertainty estimation. The idea of two classifiers is also common in segmentation, depth, and other tasks [11,12]. It, de facto, is similar to Bayesian network with two different weights. Thereby, the variance also indicates the uncertainty. More discussion can be found in "Why use an auxiliary head to estimate uncertainty?" (Line 262-272)
>
> **(2) Dropout Mechanism:** The usage of dropout is a widely adopted method for uncertainty estimation. Dropout serves as an approximation to Bayesian neural networks [10], effectively transforming a standard neural network into a Bayesian network. Bayesian networks are a common and stable approach for uncertainty estimation. As demonstrated by [10], dropout can reliably estimate uncertainty. Therefore, we have employed dropout layers in our model to leverage these benefits.
>
> For uncertainty visualization, we have visualized it in Figure 3 in our paper. The purple shows the semantic uncertainty. The green, blue, and orange show the position uncertainty of different objects. If the semantic uncertainty and positional uncertainty are high, it is reflected in the graph by darker color and wider color range, respectively. Moreover, we also visualize the uncertainty results in a rainy and a night scene in Figure 5（See Line 540-570). The colors have the same meaning as those in Figure 3. These results show that the model obtained is able to capture the uncertainties and our proposed uncertainty method can effectively help vehicles predict trajectories under night and rainy conditions respectively.
>
> [1] St-p3: End-to-end vision-based autonomous driving via spatial-temporal feature learning (ECCV 2022)
>
> [2] UniAD: Planning-oriented Autonomous Driving (CVPR 2023 Best Paper)
>
> [3] VAD: Vectorized Scene Representation for Efficient Autonomous Driving (ICCV 2023)
>
> [4] Producing and Leveraging Online Map Uncertainty in Trajectory Prediction (CVPR 2024)
>
> [5] GenAD: Generative End-to-End Autonomous Driving (ECCV 2024)
>
> [6] Accelerating Online Mapping and Behavior Prediction via Direct BEV Feature Attention (ECCV 2024)
>
> [7] Convolutional social pooling for vehicle trajectory prediction( CVPR 2018)
>
> [8] The highD dataset: A drone dataset of naturalistic vehicle trajectories on german highways for validation of highly automated driving systems (ITSC 2018)
>
> [9] The inD Dataset: A Drone Dataset of Naturalistic Road User Trajectories at German Intersections
>
> [10] Dropout as a Bayesian Approximation: Representing Model Uncertainty in Deep Learning
>
> [11] Rectifying Pseudo Label Learning via Uncertainty Estimation for Domain Adaptive Semantic Segmentation
>
> [12] Harnessing Uncertainty-aware Bounding Boxes for Unsupervised 3D Object Detection

---

> ### Author Response · Authors · 2024-11-23
> **Response to Reviewer j4Ht (Part 2)**
>
> **Q3:** What are res5c and res4f in line 94？
>
> Thanks a lot for your question. Since our backbone utilizes ResNet-50, "res5c" and "res4f" are two commonly referenced layer names within this architecture. Specifically, "res5c" refers to the output of the final layer in the fifth block of ResNet-50, while "res4f" denotes the output of the last layer in the fourth block, which precedes "res5c." We use these two terms to distinctly refer to different layers within the backbone. To clarify, we added the explanations within the manuscript (see Line 94-97).
>
> **Q4:** How did author generate Figure 1 (b)?
>
> Thank you. In the predicted map, different colors are assigned to different class categories (red for ego-vehicle, green for the boundary, blue for pedestrian crossing, orange for divider, and gray for other vehicles).
> The spatial positions (x, y) of each point are directly from the predicted μ of this point, which is 2-dimension.

---

> > ### Comment · Reviewer_j4Ht · 2024-11-25
> >
> > Thank you to the authors for their responses. I have a few follow-up comments:
> >
> > Regarding Q1:
> >
> > I believe the Argoverse dataset meets the requirements of your experiments and is widely used in the field. Conducting experiments on this dataset would greatly alleviate concerns about the generalizability of your method. I look forward to seeing these results.
> >
> > For the challenging cases, it might be more insightful to showcase how the proposed method handles failure cases from other methods. This could provide a stronger demonstration of its advantages.
> >
> > Regarding Q2:
> > I appreciate your efforts. Please ensure the additional materials are included in the revised version.
> >
> > Regarding Q3 & Q4:
> > Thank you for the clarifications.

---

> ### Author Response · Authors · 2024-11-28
> **Response to Reviewer j4Ht**
>
> Thanks for your response, here are our responses to your questions.
>
> **Q1R1：**
> We conducted an experiment on the Argoverse v2 sensor dataset, form now we can get the estimated HD map after applying our uncertainty method. Map visualization of our uncertainty method in the Argoverse v2 sensor dataset. See Figure 6 (Line 594-639).
>
> **Q1R2：**
> For the challenging cases, in Figure 4 (Line 486-580). We show the trajectory visualization results of our uncertainty method versus the no-uncertainty method when some bends/intersections are complex and have a lot of occlusion (i.e. the uncertainty of our map is large). It can be seen that when our uncertainty method is not used, the trajectory of some vehicles deviates more from GT and exceeds the road boundary range. However, after our uncertainty method is applied, the trajectory of vehicles is obviously closer to GT and does not exceed the road boundary range. The validity of our uncertainty method is proved.

---

### Official Review · Reviewer_yYh7 · 2024-11-03

**Soundness:** 2
**Presentation:** 3
**Contribution:** 2
**Rating:** 5
**Confidence:** 4

**Summary:**

The draft proposes and evaluate a new way of estimating online_HD-map uncertainties, and report a higher improvement on downstream trajectory prediction than current SoA on the topic (Gu et al. 2024).

**Strengths:**

The main interest of the paper is to pinpoint and evaluate the potential interest of using not only positional uncertainty but also "semantic" uncertainty (i.e. potential class error) of online-estimated HD-map, for improving robustness of trajectory prediction to HD-map errors.
Furthermore, according to the experiments on NuScene presented by authors, their approach can reduce the minADE and minFDE of downstream trajectory prediction by an extra ~5% compared to the recently published approach of (Gu et al. 2024).

**Weaknesses:**

There are some significant weaknesses in the draft:
  - first, contrary to what authors write, (Gu et al. 2024) do take into account also classification uncertainty (cf page 15 of that reference) ==> authors should be more rigourous on their comparison of the differences between the 2 methods
  - secondly, the way authors evaluate the positional uncertainty (lines 216-230 of the draft), i.e. by just considering 2 estimates \mu and \mu' (the second one after adding a dropout layer), seems unusual and less robust than an actual parametric estimation (such as the Laplace distribution hypothesis used by Gu et al. 2024)
 - third, authors evaluate improvements brought by their method only on variants of MapTR, but not on StreamMapNet
 - finally, the way authors present their improvement in their table 1 and text, as % reduction compared to baselines *without* (Gu et al 2024) is somewhat misleading: compared to the latter their results are only approximately -5%, rather than the highlighted -10%

**Questions:**

- on line 94 of draft, what are "res5c" and "res4f" referring to ??
- why authors do not report the improvement brought by their approach if applied to StreamMapNet ?
- given the unusual way that authors use to estimate uncertainty, some analysis of its outcome (as in section 5.2 of Gu et al. 2024) would be more than welcome

---

> ### Author Response · Authors · 2024-11-23
> **Response to Reviewer yYh7 (Part 1)**
>
> We thank Reviewer yYh7 for the insightful questions. Those questions, we believe, will further help to improve the coherence, readability, and overall quality of our paper. In the following, we have carefully addressed each of these questions and weaknesses in detail, providing additional context, supporting evidence, and refinements.
>
> **Q1:** Compare the classification uncertainty differences between the proposed method and Gu et al. 2024.
>
> Yes. Gu et al. 2024 introduce classification uncertainty, which is the classification confidence score. Our semantic uncertainty is similar to Gu et al in the target. However, it is worth noting that our semantic uncertainty design differs from that of Gu et al. in two aspects:
>
> (1) **Mechanism:** Different from directly using the confidence score in Gu et al., we leverage the discrepancy between two predictions to quantify uncertainty. Our two-head design simulates the Bayesian Networks [1, 2] in capturing the uncertainty via different model weights.
> Additionally, Gu et al. directly optimize the confidence score in the classification loss, encouraging the score convergence to 0 or 1.  The confidence score, thereby, usually can not reflect uncertainty, especially at the end of training.
> In contrast, our approach does not directly involve the confidence score, and thus is somehow robust to overfitting.
>
> (2) **Performance:** Moreover, we have added a comparison between Gu et al.'s method and our semantic uncertainty approach in Table 2, with their results highlighted in blue. Under identical map construction and trajectory prediction methodologies, incorporating our classification uncertainty yields better performance than their original classification uncertainty. Furthermore, even when using only the semantic uncertainty component (Unc_sem in Table 2) in the proposed method, our results outperform Gu et al.
>
> [1] Dropout as a Bayesian Approximation: Representing Model Uncertainty in Deep Learning (JMLR16)
>
> [2] Rectifying Pseudo Label Learning via Uncertainty Estimation for Domain Adaptive Semantic Segmentation (IJCV19)

---

> ### Author Response · Authors · 2024-11-23
> **Response to Reviewer yYh7 (Part 2)**
>
> **Q2:** The positional uncertainty seems unusual and less robust.
>
> Thank you for your question. We provide a more detailed explanation of our proposed positional uncertainty method here.
>
> (1) **Two-head Mechanism:** We use two classifiers to obtain \mu and \mu{'} for uncertainty estimation. The idea of two classifiers is also common in segmentation, depth, and other tasks [2,3]. It, de facto, is similar to Bayesian network with two different weights. Thereby, the variance also indicates the uncertainty. More discussion can be found in "Why use an auxiliary head to estimate uncertainty?" (Line 262-272).
>
> (2) **Dropout Mechanism:** The usage of dropout is a widely adopted method for uncertainty estimation. Dropout serves as an approximation to Bayesian neural networks [1], effectively transforming a standard neural network into a Bayesian network. Bayesian networks are a common and stable approach for uncertainty estimation. As demonstrated by [1], dropout can reliably estimate uncertainty. Therefore, we have employed dropout layers in our model to leverage these benefits.
>
> (3) **Performance:**  As shown in Table 2, using only the proposed positional uncertainty can surpass Gu et al. on the same HiVT trajectory prediction. If we use both positional uncertainty and semantic uncertainty, our method surpasses Gu et al. 5% minADE on HiVT and 13% minADE on DenseTNT (See Line 381-392).
>
> (4) **Analysis of Outcome:**
> (i) Uncertainty from bad weather: Figure 5 (a) (Line 540-554) depicts a rainy scene. It includes visual distortions like the cloudy weather, the reflection of the surrounding environment due to the water on the road surface, and the raindrops. These distortions will obscure the camera’s ability and introduce potential uncertainty to complicate the identification of road edges and lane markings. Through our uncertainty estimation, we can effectively identify and quantify the uncertainty in road detection under such weather conditions, enabling the vehicle to maintain the correct path. As seen from this Figure, the circle of position and semantic uncertainty of the map is larger and deeper in places where the line of sight is obscured or blurred, such as when the sidewalk is obscured by vehicles and rain. Our method has performed a good estimation and construction.
>
> (ii) uncertainty from poor visibility: Figure 5 (b) (Line 555-570) shows a night scene. Poor visibility makes it difficult for image sensors to capture accurate road information, so map estimation produces higher uncertainty regarding road locations compared to daytime scenes. Notably, at an obscured intersection hidden behind trees on the left side of the vehicle’s path, our method effectively highlights semantic uncertainty (indicated in purple) and positional uncertainty in the road and lane lines. Additionally, the positional uncertainty is significantly higher at the end of the field of view, aligning well with the expected judgment for real-world vehicle navigation.
>
> [1] Dropout as a Bayesian Approximation: Representing Model Uncertainty in Deep Learning (JMLR16)
>
> [2] Rectifying Pseudo Label Learning via Uncertainty Estimation for Domain Adaptive Semantic Segmentation
>
> [3] Harnessing Uncertainty-aware Bounding Boxes for Unsupervised 3D Object Detection

---

> ### Author Response · Authors · 2024-11-23
> **Response to Reviewer yYh7 (Part 3)**
>
> **Q3:** Add results on StreamMapNet.
>
> As suggested, we  have conducted experiments on StreamMapNet + HiVT and StreamMapNet + DenseTNT methods and obtained the following results:
>
> |                      | **HiVT**        |          |          | **DenseTNT**      |          |          |
> |---------------------------------|-----------------|----------|----------|-------------------|----------|----------|
> | **Online HD Map Method**       | minADE ↓        | minFDE ↓ | MR ↓     | minADE ↓          | minFDE ↓ | MR ↓     |
> | StreamMapNet                    | 0.3790          | 0.7822   | 0.0853   | 0.9492            | 1.740    | 0.2569   |
> | StreamMapNet + (Gu et al., 2024)| 0.3727          | 0.7492   | 0.0726   | 0.9036            | 1.645    | 0.2359   |
> | StreamMapNet + Ours             | 0.3427 (−8%)    | 0.6763 (−10%) | 0.0570 (−22%) | 0.8065 (−11%) | 1.600 (−3%) | 0.2418 (+2%) |
>
> By using StreamMapNet and combining it with our positional and semantic uncertainty for map construction, both trajectory prediction methods also achieve the best performance, especially in the HiVT method, MR metric improves by 22% than Gu et al. 2024. We also added the results and analysis in the paper (see Table 1 in blue font, Line 338-340)
>
> **Q4:** Improvement in Table 1 and text compared to baselines without Gu et al is somewhat misleading.
>
> Indeed, we do not want to overclaim our results. We will change our claims by comparing our method directly with Gu et al. instead of the baseline. When compared to Gu et al., our method achieves an improvement of approximately 5%. Consequently, we have decided to revise the comparisons in the manuscript to highlight the enhancements over Gu et al.'s approach rather than the baseline (See Line 328-340).
> Furthermore, in this specific task domain, a 5% improvement represents a significant advancement compared to Gu et al.'s method. Therefore, our contributions remain substantial and competitive in the field.
>
> **Q5:** In line 94 of the draft, what are "res5c" and "res4f" referring to?
>
> Since our backbone utilizes ResNet-50, "res5c" and "res4f" are two commonly referenced layer names within this architecture. Specifically, "res5c" refers to the output of the final layer in the fifth block of ResNet-50, while "res4f" denotes the output of the last layer in the fourth block, which precedes "res5c." We use these two terms to refer to different layers within the backbone. To clarify, we added the explanations within the manuscript (see Lines 94-97).

---

> > ### Comment · Reviewer_yYh7 · 2024-11-26
> >
> > I thank the authors for all the clarifications and details, and for addition of the new results on StreamMapNet.
> >
> > I still think that the "In contrast too" expression is at least misleading in the sentences "In contrast to the Gu et al. (2024) approach, our
> > work defines two types of uncertainty in map-based trajectory prediction tasks. The accuracy of the
> > trajectory prediction task is enhanced by addressing the issue of map noise through positional and
> > semantic uncertainty. " (lines 142-145). Authors still seem to consider that a difference of their work from that of Gu et al. (2024) is the use of both positional and semantic uncertainty. Which is false (cf page 15 of that reference) ; what IS original is authors' different way of handling semantic uncertainty (BTW, I suscribe to their remark that after minimizing NLL, the scalar scores for classes tend to become nearly binary thresholding rather than probabilities).
> >
> > I think authors should rather focus their paper on that difference, and provide empirical evidence comparing the stability and effectiveness of their method to parametric approaches like the Laplace distribution. I suggest that authors should establish a detailed comparison table highlighting the specific differences in how their method and Gu et al.'s approach handle semantic/classification uncertainty. It would also be quite desirable to include a subsection similar to Section 5.2 in Gu et al. (2024), specifically analyzing how their uncertainty estimates correlate with actual prediction errors or map inaccuracies.
> >
> > The evaluation results are clearly in favor of the proposed approach. But it would be desirable to at least roughly understand why.
> > In summary I encourage authors to take more time than just a rebuttal period, in order to prepare a more solid and detailed paper that would not only highlight their good numerical results, but also *explain them* by better comparison with Gu et al. (2024), and even possibly demonstrate that their approach id mathematically better-founded ?
> > I therefore stay on my rating of 5, to be considered as an incitation to prepare a much better version of their paper that the presented work probably deserves.

---

> ### Author Response · Authors · 2024-11-28
> **Response to Reviewer yYh7**
>
> Thank you for your reply. Here are our replies to your questions:
>
> 1.  Some misleading expressions
>
> We do not intend to mislead or overclaim.  We have replaced "In contrast to" with "different from" in the updated pdf (Line 142).
> Yes.  We focus more on better uncertainty formulation. The revised sentence is as our discussion.
> Different from directly using the confidence score as uncertainty in Gu et al., we leverage the discrepancy between two predictions to quantify uncertainty. Our two-head design simulates the Bayesian Networks in capturing the uncertainty via different model weights. Additionally, Gu et al. directly optimize the confidence score in the classification loss, encouraging the score convergence to 0 or 1. The confidence score, thereby, usually can not reflect uncertainty, especially at the end of training. In contrast, our approach does not directly involve the confidence score and thus is somehow robust to overfitting.
>
> 2. Laplace distribution and more detailed comparison with Gu et al.
>
> Thank you.
> 	We have added the mathematical proof of our uncertainty method in detail in the appendix of the paper (Line 883-988).
> Our method also applied the Laplace distribution for both mu and mu' (Line 215).
>
> We will make the illustration clear. Laplace distribution makes the distribution sharper, thereby discriminating certain and uncertain samples.
> "Following Gu et al, we apply the Laplace distribution to make the distribution sharper, thereby discriminating the certain and uncertain samples".
>
> 3. Comparison with Section 5.2 in Gu et al.
>
> Yes. We have tested it in the updated paper. Please check Section 4.3 (Line 477-565 and 577-583).
>
> 4. Thanks a lot.
>
> We really appreciate your help. We are trying our best to compare Gu et al in a fair way. If any experiments are needed, we are happy to provide as much as possible. After 27 Nov, we may not update the pdf but still reply to you with the latest result.

---

### Official Review · Reviewer_SyAv · 2024-11-04

**Soundness:** 3
**Presentation:** 4
**Contribution:** 3
**Rating:** 6
**Confidence:** 4

**Summary:**

This article introduces a novel framework for trajectory prediction that addresses the challenges posed by real-time map noise resulting from sensor inaccuracies, misinterpretations, and other factors. The proposed method estimates both positional and semantic uncertainties, integrating them into the prediction process through a dual-head structure that performs independent predictions and extracts prediction variance as an indicator of uncertainty. Validated on the nuScenes dataset, this approach effectively captures map noise and enhances existing trajectory prediction methods.

**Strengths:**

1. The writing in this article is excellent, featuring clear logic and organization. Additionally, the figures and charts adhere to established standards, which is commendable.

2. This paper addresses two types of uncertainties: positional and semantic. It aims to mitigate the adverse effects of noise in High-Definition (HD) map estimation for trajectory prediction.

3. By integrating map elements that include both positional and semantic uncertainties into downstream models, this approach leverages the context of uncertainty to improve the accuracy of trajectory predictions.

4. The method can be integrated with existing mapping estimation and trajectory prediction approaches, consistently enhancing prediction accuracy. This is evidenced by significant improvements in minADE, minFDE, and MR on the nuScenes dataset when combined with specific backbone models.

**Weaknesses:**

I have the following concerns:

1. Although the authors have conducted numerous experiments and analyses, which is commendable, there is still an opportunity to expand the experiments further to enhance the credibility of the work. For instance, the authors could consider referencing some of the latest research in online mapping or trajectory prediction, such as:

    [1] MapTracker: Tracking with Strided Memory Fusion for Consistent Vector HD Mapping

    [2] MGMap: Mask-Guided Learning for Online Vectorized HD Map Construction

    [3] MapDistill: Boosting Efficient Camera-based HD Map Construction via Camera-LiDAR Fusion Model Distillation

    [4] Producing and Leveraging Online Map Uncertainty in Trajectory Prediction

    [5] Accelerating Online Mapping and Behavior Prediction via Direct BEV Feature Attention

2. In future work, it is suggested that the authors attempt to explore research based on scene topology reasoning, such as:
    [1] TopoNet: A New Baseline for Scene Topology Reasoning

**Questions:**

As stated in the Weakness.

---

> ### Author Response · Authors · 2024-11-23
> **Response to Reviewer SyAv**
>
> We sincerely appreciate the constructive opinions from Reviewer SyAv. We thank the positive feedback, for example, the writing is excellent, featuring clear logic and organization. We believe the proposed suggestions, for example, discussions with some related works will further improve the quality of our paper. We will address these questions and weaknesses in more detail.
>
> **Q1:** Consider referencing some of the latest research in online mapping or trajectory prediction.
>
> Thanks a lot for the great suggestions. We have discussed three map construction papers, and updated the "Online Map Estimation" section of the related works (Line 163-174). The fourth paper [4] serves as our comparative method and baseline, which we have discussed in the related works and evaluated in subsequent performance analysis (Line 134-135 and Table 1). The fifth paper further improves trajectory prediction by considering BEV features and is the latest work of this task. We added the discussion in the "Map-informed Trajectory Prediction" of the related works (Line 137-140).
>
> **Q2:** Explore research based on scene topology reasoning in future work.
>
> We appreciate your great suggestion. Unlike traditional lane detection or map construction methods, TopoNet infers the connectivity between lane centerlines and various traffic elements from sensor inputs. It specifically leverages a graph instance feature transmission neural network to extract prominent predictive cues from other elements within the topological map. In contrast, our trajectory prediction approach focuses on constructed vectorized maps along with the uncertainty to forecast vehicle trajectories. We add the discussion of TopoNet in the related works (Line 135-136).

---

> ### Author Response · Authors · 2024-11-28
> **Response to Reviewer SyAv**
>
> We thank you again.
>
> In order to prove the generalization and validity of our method, we added the following experiments:
>
> 1. Add our method to the visual HD map after applying our uncertainty method on the dataset Argoverse v2 sensor dataset, refer to Figure 6 (Line 594-638). In addition, the visual analysis of some complex scenes on the nuScenes dataset is added. See Figure 5 (Line 540-565).
>
> 2. Three new trajectory prediction methods are added [1,2,3]. The trajectory prediction performance is improved after the introduction of map information after our uncertainty estimation, as shown in Table 3 (Line 567-574).
>
> [1] Convolutional social pooling for vehicle trajectory prediction, CVPR 2018.
>
> [2] Leapfrog diffusion model for stochastic trajectory prediction, CVPR 2023.
>
> [3] Wsip: Wave superposition inspired pooling for dynamic interactions-aware trajectory prediction, AAAI 2023.

---

### Meta-Review · Area_Chair_W32x · 2024-12-22

**Metareview:**

This work proposes a novel trajectory prediction approach that uses both, positional and semantic uncertainty. It demonstrates strong empirical results outperforming previous work on using map uncertainty for trajectory prediction and makes a strong case for its design choices. The main remaining concern is the novelty of the work. Previous works have already established that uncertainty can be useful, and thus, the main technical contribution here seems to be how uncertainty is estimated. While integrating a different uncertainty estimation technique for better results is novel, the novelty is small in that optimizing the type of uncertainty with established techniques would be expected to lead to better results. Investigating uncertainty at the intersection of online mapping and driving remains a relevant research topic, and it would be interesting to see a broader analysis of which uncertainty estimation techniques are particularly suitable for this task. This, however, is much broader and beyond the scope of what has been submitted in the present work.

**Additional Comments On Reviewer Discussion:**

The authors addressed most of the reviewer's comments, with the point about novelty remaining the least satisfactory in their response.

---

### Decision · Program_Chairs · 2025-01-22

Reject